# Balancing time use for children's fitness and adiposity: Evidence to inform 24-hour guidelines for sleep, sedentary time and physical activity

Dorothea Dumuid[ID][1]*, Melissa Wake[2,3], David Burgner[2,3], Mark S. Tremblay[4], Anthony D. Okely[5,6], Ben Edwards[7], Terence Dwyer[2,3,8,9], Timothy Olds[1]

1 Alliance for Research in Exercise, Nutrition and Activity (ARENA), Allied Health & Human Performance, University of South Australia, Adelaide, South Australia, Australia, 2 Murdoch Children's Research Institute, Parkville, Victoria, Australia, 3 Department of Paediatrics, The University of Melbourne, Parkville, Victoria, Australia, 4 Children's Hospital of Eastern Ontario Research Institute and Department of Pediatrics, University of Ottawa, Ottawa, Ontario, Canada, 5 Early Start, Faculty of Social Sciences, University of Wollongong, Wollongong, New South Wales, Australia, 6 Illawarra Health and Medical Research Institute, Wollongong, New South Wales, Australia, 7 Centre for Social Research and Methods, Australian National University, Canberra, Australian Capital Territory, Australia, 8 Oxford Martin School, University of Oxford, Oxford, United Kingdom, 9 Nuffield Department of Women's & Reproductive Health, University of Oxford, Oxford, United Kingdom

* dot.dumuid@unisa.edu.au

**Data Availability Statement:** We are not the owners of the CheckPoint and LSAC data. There are legal restrictions on public sharing of the data.

## Abstract

### Purpose

Daily time spent on one activity cannot change without compensatory changes in others, which themselves may impact on health outcomes. Optimal daily activity combinations may differ across outcomes. We estimated optimal daily activity durations for the highest fitness and lowest adiposity.

### Methods

Cross-sectional Child Health CheckPoint data (1182 11-12-year-olds; 51% boys) from the population-based Longitudinal Study of Australian Children were used. Daily activity composition (sleep, sedentary time, light physical activity [LPA], moderate-to-vigorous physical activity [MVPA]) was from 8-day, 24-hour accelerometry. We created composite outcomes for fitness (VO$_{2max}$; standing long jump) and adiposity (waist-to-height ratio; body mass index; fat-to-fat-free log-ratio). Adjusted compositional models regressed activity log-ratios against each outcome. Best activity compositions (*optimal time-use zones*) were plotted in quaternary tetrahedrons; the *overall optimal time-use composition* was the center of the overlapping area.

### Results

Time-use composition was associated with fitness and adiposity (all measures p<0.001). Optimal time use differed for fitness and adiposity. While both maximized MVPA and

To become a licensed data user for the CheckPoint and LSAC studies, researchers are required to sign a confidentiality deed, found here: https://ada.edu.au/confidentiality-deed-poll-november-2019/. The confidentiality deed imposes a legal obligation to not share any unit-level data with anyone unless they also are registered data users. Researchers can apply to become registered data users by following the link provided: https://dataverse.ada.edu.au/dataverse/ncld. All the data used in our study can be downloaded at this website, except we used a slightly different form of accelerometry data. The accelerometry data currently available in the CheckPoint dataset have all minutes when the accelerometers were removed for sport classified as MVPA (moderate-to-vigorous physical activity). For our study, we requested CheckPoint to provide us accelerometry data where the sport non-wear time was replaced with 50% MVPA, 30% light physical activity and 20% sedentary time, as we believe this better reflects children's activity levels during sport. The alternate accelerometry files are available to registered users, upon request to the CheckPoint team at LSAC.ChildHealthCheckPoint@mcri.edu.au.

**Funding:** DD was supported by the National Health and Medical Research Council (NHMRC) [Early Career Fellowship 1162166] and the National Heart Foundation of Australia [Postgraduate Fellowship 102084]. MW was supported by the NHMRC [Principal Research Fellowship 1160906]. DB was supported by an NHMRC Investigator Grant [1175744]. ADO was supported by an NHMRC Leadership Fellowship [1176858]. The Child Health CheckPoint was supported by the NHMRC of Australia [Project Grants 1041352, 1109355], The Royal Children's Hospital Foundation [2014-241], the Murdoch Children's Research Institute (MCRI), The University of Melbourne, the National Heart Foundation of Australia [100660] and the Financial Markets Foundation for Children [2014-055, 2016-310]. Research at the MCRI is supported by the Victorian Government's Operational Infrastructure Support Program. The funders had no role in study design, data collection and analysis, decision to publish, or preparation of the manuscript.

**Competing interests:** The authors have declared that no competing interests exist.

minimized sedentary time, optimal *fitness* days had higher LPA (3.4 h) and shorter sleep (8.25 h), but optimal *adiposity* days had lower LPA (1.0 h) and longer sleep (10.9 h). Balancing both outcomes, the overall optimal time-use composition was (mean [range]): 10.2 [9.5; 10.5] h sleep, 9.9 [8.8; 11.2] h sedentary time, 2.4 [1.8; 3.2] h LPA and 1.5 [1.5; 1.5] h MVPA.

## Conclusion

Optimal time use for children's fitness and adiposity involves trade-offs. To best balance *both* outcomes, estimated activity durations for sleep and LPA align with, but for MVPA exceed, 24-h guidelines.

## Introduction

Children's fitness and adiposity are related to how they spend their time in various activities. Fitness is likely to improve when more time is spent in activities of higher intensity (moderate and vigorous), which load the cardiorespiratory system and provide opportunity to build muscle strength and power [1]. Conversely, fitness may decline when less time is spent in such activities, in favor of sedentary behaviors and sleep. Similarly, adiposity may decrease when children spend more time in moderate-and-vigorous intensity activities because their energy expenditure increases. Getting sufficient sleep may also lead to decreased adiposity due to better regulation of stress hormones which conserve energy [2]. It may be possible that too much sedentary and light activity could rob time from higher intensity activities, increasing adiposity.

Parents and caregivers want to know how much time their children should spend in daily activities, and public health guidelines attempt to provide recommended daily durations of sleep, screen time and physical activity [3]. However, the main body of existing evidence does not address these questions. It simplistically describes how *more* or *less* of an activity is associated with an outcome, without providing an indication of *how much* daily time should be devoted to an activity.

This is because most studies use linear regression to explore how various daily activities are associated with fitness and adiposity. From these models, it is clear that more time in moderate-to-vigorous physical activity (MVPA) and less time sedentary are consistently associated with better fitness and adiposity [1, 4]. The relationships between time spent in light physical activity (LPA) and fitness and adiposity are less clear and inconsistent [1]. Longer sleep duration is beneficially associated with adiposity [5], but its association with fitness is less well understood.

Several studies have used alternative analytical methods, such as splines or generalized additive models, to determine whether there may be an optimal duration of activities [6–8]. However, these studies did not take into account that all daily activities are interrelated because they compete for time-shares within a finite 24-hour window. If more time is spent in one activity, time must be taken from one or more of the remaining activities to maintain the fixed total of 24 hours [9]—which itself may impact on outcomes. Accordingly, analyses should explore the optimal duration of activities across a day in relation to health outcomes. The best durations for one health outcome may not the best durations for another, creating a variety of competing demands.

All daily activities can be analyzed simultaneously in the same model using compositional data analysis (CoDA) [10, 11]. An increasing number of CoDA studies are exploring the relationships between daily activities and health outcomes [12]. In general, these studies have identified that higher MVPA, relative to the other daily activities, is beneficially associated with both children's fitness and adiposity, while longer sleep, shorter sedentary time and shorter LPA (each relative to other activities) is associated with lower adiposity only [13–17]. In a large, cross-sectional population-based sample, we aimed to apply CoDA in a novel way to determine similarities and differences in the optimal daily combinations of daily activities for two important aspects of children's health: fitness and adiposity. We also aimed to determine a trade-off combination of daily durations of activities that optimizes both outcomes. By determining the trade-off combination we aimed to describe the Goldilocks Day [18, 19], where activity durations are "just right" for both fitness and adiposity.

## Materials and methods

### Study design and participants

Participants were from the Child Health CheckPoint [20, 21], a cross-sectional module nested between Waves 6 and 7 of the population-based Longitudinal Study of Australian Children (LSAC) [22]. All children in LSAC Wave 6 (n = 3764) were invited to participate in CheckPoint, without exclusion. A total of 1874 participated in CheckPoint, and of these, 1279 had valid and complete accelerometry data. Participants with complete accelerometry, outcome and covariate data were included in this study (n = 1182 for adiposity and n = 1137 for fitness). Fig 1 shows the participant flow. A parent/guardian provided written informed consent for their child's participation. Ethical approval was granted by The Royal Children's Hospital (Melbourne) Human Research Ethics Committee (HREC33225) and the Australian Institute of Family Studies Ethics Committee (AIFS14-26).

### Measurements

Measurements were taken by trained staff at a CheckPoint Assessment Centre in one of Australia's seven major cities or at a Mini Centre in one of eight regional cities. Participants unable to attend a center were offered a home visit.

**Exposure: Daily activity composition.** At completion of the visit, a research assistant fitted the participants' non-dominant wrist with a GENEActiv accelerometer (Activinsights Ltd., UK), and asked them to wear it 24 hours a day, for 8 days [23]. Participants recorded bed and wake times, and time/reason for accelerometer removal on a paper-based log. Accelerometry data, collected at 50 Hz, were downloaded using GENEActiv PC Software (Activinsights, UK), and converted to 60-second epoch files. A MATLAB-based customized software program (*Cobra)*, the algorithm by van Hees et al [24], visual inspection of accelerometer traces and the paper-based logs were used to identify sleep and non-wear time. The 60-second epochs were classified as sleep if they contained equal or more sleep vs wake 5-second epochs. If the device was removed for "sport", the corresponding period of non-wear was imputed with 50% MVPA, 30% LPA and 20% sedentary time [25].

Validated cut points for GENEActiv devices in school-aged children were used to classify the 60-s epochs into intensity bands [26]. Cut points were linearly adjusted for the 50 Hz sampling frequency to 244 gravity minutes (g.min, i.e., acceleration because of gravity multiplied by minutes) for sedentary time, 878 g.min for LPA and 2175 g.min for MVPA. Accelerometer days were considered invalid if waking wear time was ≤10 h, or if average sleep was ≤200 min/d or sedentary time ≥1000 min/d. Participants were required to have at least four valid days.

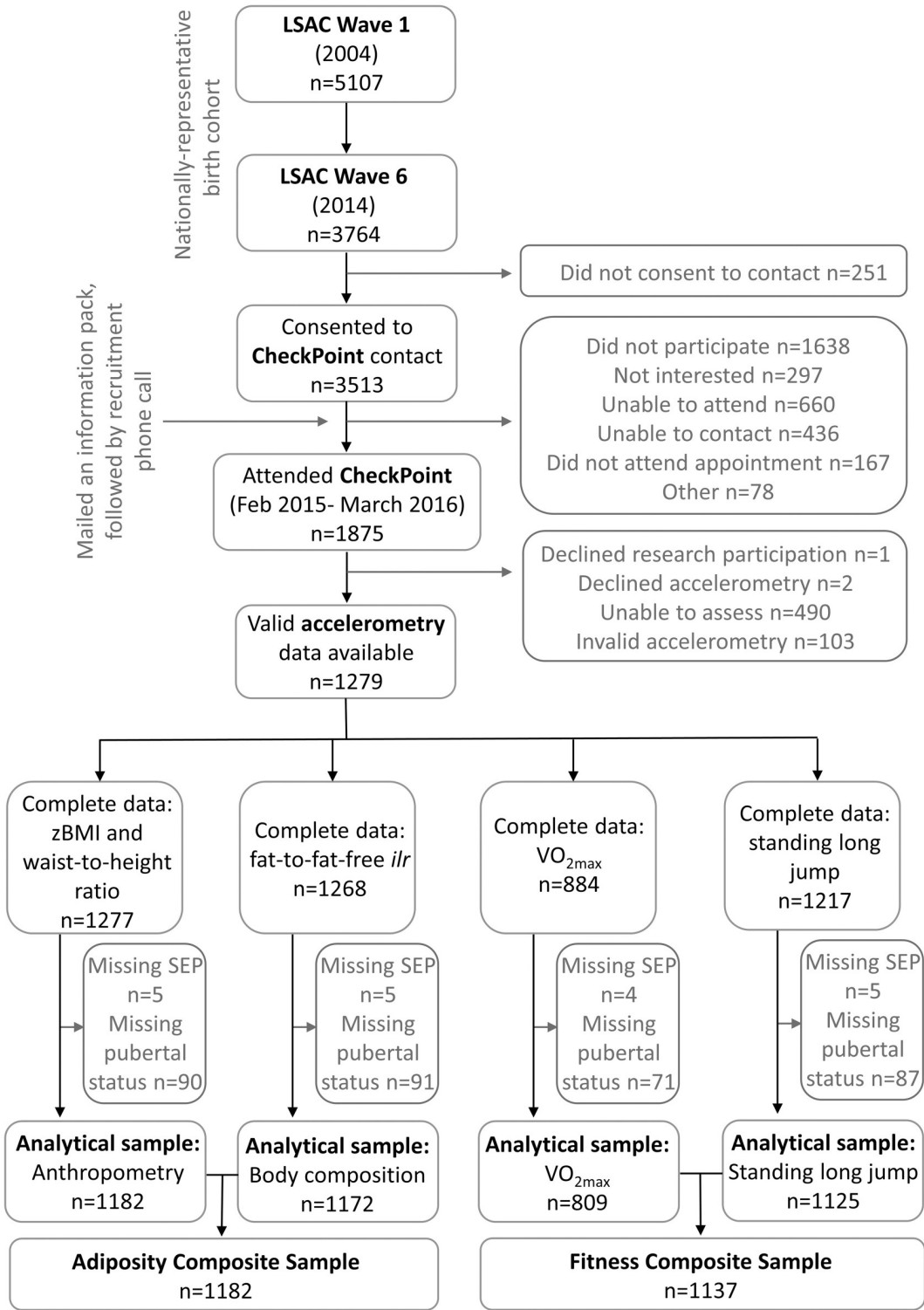

**Fig 1. Participant flow.** LSAC = Longitudinal Study of Australian Children; *z*BMI = body mass index *z*-score; *ilr* = isometric log ratio; $VO_{2max}$ = predicted maximal aerobic power; SEP = socioeconomic position.

**Outcomes: Fitness.** Maximal aerobic power (VO₂max) was estimated from the PWC170 test [27] using a cycle ergometer (Monark 928G3, Sweden) and heart rate monitor (Polar FT4, Finland). Following a two-minute warm up, the test involved three to four two-minute bouts. Each bout had a higher work rate than the last. Increases in work rate were dependent on participant's heart rate measured in the last 15 seconds of each bout. Estimated maximal work rate was calculated from a regression of work rate on heart rate for each stage, extrapolated to an estimated maximal heart ($HR_{max}$) [28]. $VO_{2max}$ was estimated from maximal work rate ($WR_{max}$):

$$VO_2 max \, (\text{ml/kg/min}) \; = \; (0.012 \, WRmax \, (\text{W}) \; + \; 0.36)/Body \, mass \, (\text{kg}).$$

[17].

Standing long jump required a two-footed take-off and participants were encouraged to swing their arms and bend their knees. Distance was measured from the starting tape on the mat to the hindmost part of the rear foot on landing. The greatest distance in centimetres from three attempts was recorded.

A composite fitness score was created by expressing $VO_{2max}$ and long jump distance as sample-specific *z*-scores. Available *z*-scores were averaged, with a higher score representing better fitness.

**Outcomes: Adiposity.** Body composition and weight were measured in semi-fasted condition, using the InBody230 four-limb segmental body composition bioelectric impedance scale (Biospace, Seoul, Korea), in light clothing without shoes or socks [29]. Bioelectric impedance is a valid (r = 0.69–0.79 vs underwater weighing) and reliable ($CV_{intra}$ = 3%) method for estimating body fat in school-aged children [30]. Body composition was expressed as the isometric log ratio *(ilr)* of fat vs fat-free mass, $\frac{1}{\sqrt{2}} \left( \frac{Fat \, mass}{Fat-free \, mass} \right)$ to respect the compositional nature of the variable [31]. Height was measured with a stadiometer (Invicta I0955, Leicester, UK). Body mass index (BMI) was calculated as weight (kg)/height (m)$^2$ and expressed as age- and sex-specific *z*-scores [32]. Waist circumference was measured with a steel anthropometric tape (Lufkin W606PM) midway between the bottom of the 10th rib and the top of the iliac crest in the mid-axillary line [33].

A composite adiposity summary score was created by expressing the body composition *ilr*, *z*BMI and waist-to-height ratio as sample-specific *z*-scores. Available *z*-scores were averaged, and the composite score inverted so that higher values represented more favourable adiposity.

**Covariates.** Models were adjusted for sex, age, pubertal development and socioeconomic position. Fitness models were aditionally adjusted for zBMI. Participants' sex and age were obtained from LSAC. Pubertal signs were self-reported using an iPad interface of the Pubertal Development Scale, enabling classification into either pre-pubertal, early pubertal, mid-pubertal, late pubertal or post-pubertal [34]. For analysis, pubertal development was treated as a continuous variable. Family-level socioeconomic position (SEP) was obtained from Wave 6 of LSAC, which releases an internal composite *z*-score derived from parental occupation, education and income with each wave [35].

## Data treatment and analysis

Average daily time-use composition (sleep, sedentary time, LPA, MVPA) was weighted at 5: 2 for weekday: weekend day. The mean of measured time use was 1432 min/d (SD = 23).

Zeros in MVPA for two participants were replaced by 65% of smallest possible non-zero value to enable log-ratio transformations to be applied for CoDA [36]. Time-use compositions were described by their compositional center (geometric means of activities, linearly adjusted

to sum to 1440 min/d). Nonparametric bootstrap 95% CI were calculated using 1000 replicates.

**Relationship between time-use composition and outcomes.** Detailed description of the modelling is provided in S1 File. Time-use compositions were expressed as *ilrs* using *Compositions* [37] in R (R Development Core Team, 2008). Robust multiple linear models regressed each of the individual and composite fitness and adiposity outcomes against time-use *ilrs*, adjusted for covariates. Quadratic terms for the *ilrs* were retained if they improved the model fit (partial F test for nested models $p<0.05$). Fitness models were additionally adjusted for *z*BMI. Variance inflation factors for exposure variables were below acceptable limits in all models (all $<1.7$) [38].

We used the above models as predictive formulas to estimate the individual and composite outcomes for every possible combination of daily activities (in 10-minute increments) within the limits observed within the study sample (truncated at ±3 SD of their univariate distributions). There were 6330 predictive compositions ranging between: sleep = 7.1 to 11.7 h; sedentary = 7.8 to 14.7 h; LPA = 0.7 to 5.0 h; and MVPA = 0.2 to 1.5 h. Estimated outcomes were plotted against incrementally increasing min/d for each individual activity.

**Optimal time-use zones.** For each individual and composite outcome, predictive time-use compositions were ordered from best to worst. Compositions associated with the best levels of the outcomes (best 5%, *optimal time-use zones*) were described by means and range. To find the best overall "compromise" time-use zone considering both fitness and adiposity, we determined where the best time-use zones for fitness and adiposity composite scores overlapped. The outer limits of the overlapping "compromise" time-use zone were described for each activity dimension. We identified the compositional center of the overlapping zone as the *overall optimal time-use composition* for fitness and adiposity. The zones were illustrated with quaternary plots using the plot3D function from the *rgl* package [39].

## Results

Children (n = 1182, 63% of CheckPoint sample) were included in the analyses if they had valid accelerometry data and complete covariate and outcome data (Fig 1). Compared to children in the original population-representative LSAC B cohort without valid/complete CheckPoint accelerometry and covariate data, those with these data (and therefore in these analyses) had higher household-level SEP (SEP *z*-score mean 0.24, SD 0.99 vs. mean -0.11, SD 0.98) (Table 1). On average they also had higher neighborhood advantage, according to a census-derived composite index of relative socio-economic disadvantage at postcode level (SEIFA) [40], with slightly higher advantage and homogeneity (mean 1028, SD 60) than Australians in general (national mean 1000, SD 100).

### Relationships between activity composition and fitness and adiposity

The set of activity composition *ilrs* was associated with each of the fitness and adiposity outcomes, as well as the composite fitness and adiposity scores (all $p<0.001$) (Table 2). It is striking that the best zone mean for fitness comprised around 2.5 hours less in sleep and 2.5hrs more in LPA than the best zone mean for adiposity, with a much smaller mean excess in MVPA of 13 min.

Fitted loess curves depict higher MVPA was associated with better fitness and adiposity (Fig 2 and S2 File). The MVPA response curves showed a clear logarithmic pattern. The adiposity sedentary time curve appeared relatively flat, with a slight negative relationship. On average the fitness sedentary curve showed a negative relationship for durations >570 min/d.

**Table 1. Descriptive statistics.**

| Characteristic | | Value (n = 1182) |
|---|---|---|
| Age (mean (SD)) | Years | 12.0 (0.4) |
| Sex; n (%) | Male | 603 (51) |
| | Female | 579 (49) |
| SEP *z*-score (mean (SD)) | | 0.24 (0.99) |
| Puberty; n (%) | Pre-pubertal | 121 (10) |
| | Early pubertal | 298 (25) |
| | Mid-pubertal | 605 (51) |
| | Late pubertal | 152 (13) |
| | Post-pubertal | 6 (1) |
| Activity behaviors: arithmetic mean (SD) | Sleep, min/d | 566 (47) |
| | Sedentary, min/d | 675 (72) |
| | Light physical activity, min/d | 164 (46) |
| | MVPA, min/d | 28 (22) |
| Activity behaviors: compositional mean[a] (bootstrapped 95% CI) | Sleep, min/d | 575 (573; 578) |
| | Sedentary, min/d | 685 (680; 689) |
| | Light physical activity, min/d | 160 (158; 163) |
| | MVPA, min/d | 20 (19; 21) |
| Fitness (mean (SD)) | $VO_{2max}$, ml/kg/min | 47.9 (9.8)[n=809] |
| | Standing long jump, cm | 139 (22)[n=1125] |
| Adiposity (mean (SD)) | *z*BMI | 0.4 (1.1) |
| | Waist, cm | 66 (8) |
| | Height, cm | 154 (8) |
| | Waist-to-height ratio | 0.43 (0.05) |
| | Fat mass, kg | 10.4 (6.4)[n=1172] |
| | Fat free mass, kg | 35.1 (5.8)[n=1172] |
| | %Body fat | 21.5 (8.4)[n=1172] |
| | *ilr*(fat: fat free) | -0.97 (0.35)[n=1172] |

SEP = socioeconomic position; MVPA = moderate-to-vigorous physical activity; $VO_{2max}$ = predicted maximal aerobic power; BMI = body mass index; *ilr* = isometric log ratio: $\frac{1}{\sqrt{2}} \ln\left(\frac{Fat\ mass}{Fat\ free\ mass}\right)$.

[a]Compositional mean was calculated by finding the geometric mean of each activity and linearly adjusting these values to collectively sum to 1440 (min/day).

Notably, higher sleep was associated with worse fitness but better adiposity, while higher light physical activity was associated with better fitness and worse adiposity.

## Optimal daily composition for fitness and adiposity

The center (compositional mean) and range [min; max] of the set of predictive compositions associated with the optimal (best 5%, or 95[th] percentile) fitness and adiposity measures and summary scores are presented in Table 2.

Overall optimal time use (best "compromise" between fitness and adiposity outcomes) was conceptualized as the center of the region where the best fitness and adiposity composite time-use zones overlapped (Fig 3, see S3 File for interactive 3-D plot). As the time-use zones representing the best 5% and 10% of estimated fitness and adiposity composites did not overlap, we used the best 15%. The center [range] of the overlapping zone was: 10.2 [9.5; 10.5] h sleep; 9.9

**Table 2. Relationships between activity composition and outcomes: Type III ANOVA model summaries and estimated optimal activity zones for fitness and adiposity outcomes.**

| Measure | Model Summary for Activity Composition[a] | | | Optimal zone[b]: mean [range] (h/d) | | | |
|---|---|---|---|---|---|---|---|
| | n | F | p | Sleep | Sedentary | LPA | MVPA |
| Standing Broad Jump[cd] | 1125 | 11.5 | <0.001 | 8.3 [7.2; 9.5] | 11.3 [9.3; 13.5] | 3.0 [1.7; 4.7] | 1.4 [1.3; 1.5] |
| $VO_{2max}$[cd] | 809 | 11.0 | <0.001 | 8.0 [7.2; 9.3] | 10.6 [8.7; 12.7] | 4.0 [2.7; 5.0] | 1.4 [1.2; 1.5] |
| BMIz | 1182 | 14.2 | <0.001 | 10.9 [8.5; 11.7] | 11.1 [8.7; 13.3] | 0.9 [0.7; 2.2] | 1.1 [0.3; 1.5] |
| Waist-to-height ratio | 1182 | 5.8 | <0.001 | 10.7 [8.2; 11.7] | 11.2 [8.8; 13.7] | 0.9 [0.7; 2.0] | 1.2 [0.5; 1.5] |
| Fat-to-fat-free mass *ilr*[c] | 1172 | 11.2 | <0.001 | 10.5 [7.2; 11.7] | 10.7 [8.8; 12.0] | 1.5 [0.7; 5.0] | 1.3 [0.3; 1.5] |
| Fitness Composite[cd] | 1137 | 17.0 | <0.001 | 8.3 [7.2; 9.7] | 10.9 [9.2; 12.8] | 3.4 [2,2; 5.0] | 1.5 [1.3; 1.5] |
| Adiposity Composite | 1182 | 14.9 | <0.001 | 10.9 [8.3; 11.7] | 10.8 [7.8; 13.5] | 1.0 [0.7; 3.0] | 1.2 [0.5; 1.5] |

Fitness composite includes $VO_{2max}$ and standing long jump distance. Adiposity composite includes *z*BMI, waist-to-height ratio and fat-to-fat-free mass *ilr*. LPA = light physical activity; MVPA = moderate-to-vigorous physical activity. *F* = multiple regression coefficient for the set of activity composition *ilr*s, adjusted for age, sex, puberty and socioeconomic position. $VO_{2max}$ = predicted maximal aerobic power, *z*BMI = body mass index *z*-score, *ilr* = isometric log ratio.

[a]the set of *ilr*s.

[b]Optimal zone refers to predictive activity compositions associated with the best 5% (95th percentile) of the outcome measure.

[c]Models include quadratic term for the activity composition *ilr*s.

[d]Models additionally adjusted for *z*BMI

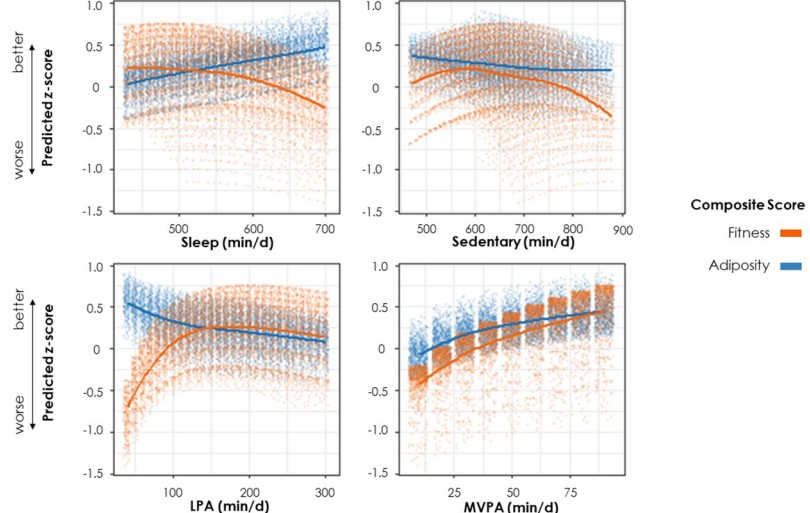

**Fig 2. Relationship between incrementally increasing durations of individual activity behaviors and fitness/adiposity composite summary *z*-score.** Higher *z*-scores represent better outcomes. Adjusted for age, sex, puberty and socioeconomic position. Fitness additionally adjusted for body mass index *z*-score and included quadratic term for the activity composition. Note: Jitter was applied to data points to enable visualization of overlapping points. Each data point represents one of the possible permutations of activity compositions (in 10-min increments) within the study sample's empirical activity footprint (i.e., the ranges of activity durations observed in the sample truncated at ±3SD of univariate distributions of behaviors). Care must be taken when interpreting the relationship between individual activity behaviors and outcomes. Although we can describe the shape of relationships in terms of individual behaviors (e.g., MVPA is beneficially associated with outcomes), this description pertains to the average situation only, as shown by the loess line. There is substantial variation around this line because the observed relationship with the activity in question (e.g., MVPA) depends on the values of the remaining activity behaviors (e.g., sleep, sedentary time and LPA). LPA = light physical activity; MVPA = moderate-to-vigorous physical activity.

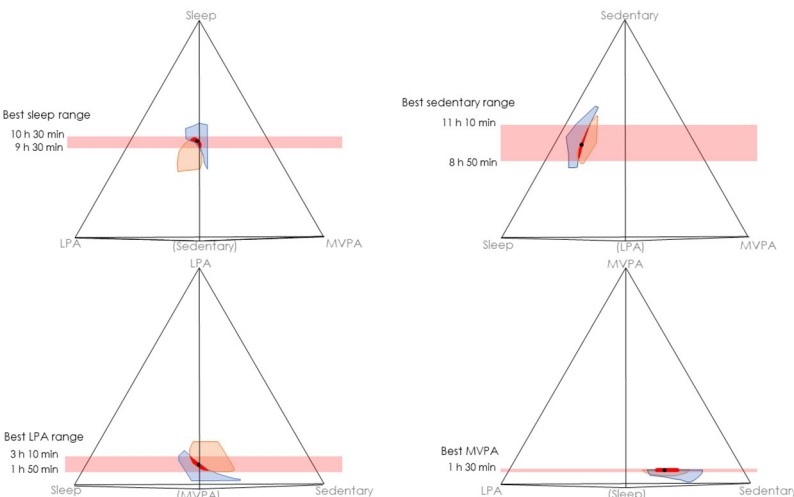

**Fig 3. Estimated best zones for fitness (orange) and adiposity (blue) corresponding to time-use compositions associated with the best 15% of each outcome.** Red denotes the best overall zone, where fitness and adiposity best zones overlap. Models adjusted for sex, age, puberty and household socioeconomic status. Fitness models additionally adjusted for body mass index *z*-score and included a quadratic term for the activity composition. The four plots show different rotations/perspectives of the same 3-D tetrahedron. Activities are at 100% (24 hours) at the corresponding apices of the tetrahedron, and 0% at the opposite base. A datapoint in the exact center of the tetrahedron would have equal shares of each activity (25%, or 6 hours). The activity in brackets is in the foreground. For an interactive 3-D plot, please see S3 File. The compositional mean of the overlap zone between the polygons (shown in red) is indicated by the black dot (h/day): Sleep = 10.2; Sedentary = 9.9; LPA = 2.4; MVPA = 1.5. LPA = light physical activity, MVPA = moderate-to-vigorous physical activity.

[8.8; 11.2] h sedentary time; 2.4 [1.8; 3.2] h LPA; and 1.5 [1.5; 1.5] h MVPA (in the overlapping zone all MVPA = 1.5 h).

## Discussion

### Principal findings

The daily time use of 11-12-year old children was significantly associated with fitness and adiposity outcomes. The optimal day for fitness had similar MVPA and sedentary time to the optimal day for adiposity but differed strikingly in sleep (+2 h 39 min) and LPA (- 2 h 24 min). Based on the activity classifications used in this study, the best trade-off day to optimize both outcomes was estimated to be 10.2 h sleep, 9.9 h sedentary time, 2.4 h LPA and 1.5 h MVPA.

### Comparison with previous literature

Previous CoDA studies have reported beneficial associations of more MVPA and less sedentary time (relative to one or more other activities) with fitness and adiposity [13, 14, 17], and more sleep and less LPA (relative to other activities) with adiposity [11, 14].

Previous studies have reported curvilinear relationships between more self-reported physical activity and better fitness [41] or, using CoDA, between more MVPA and lower adiposity [11, 16], with the steepest association in the first 15- to 30-min/d as in our study. Although plateauing, the benefits of MVPA appear to continue to increase with increasing duration; lack of a "U-shape" may reflect the rather small maximal amount of MVPA that children naturally attain. Our estimate of optimal daily MVPA (1.5 h) is higher than international 24-h movement guidelines of "at least 60 minutes" [42–44], although they do suggest that longer durations may be more beneficial. Studies similarly examining LPA appear to be lacking, however

current movement guidelines of "several hours per day" appear broadly consistent with our estimated optimal duration (2.4 h).

Unlike other studies [45, 46], we did not see U-shaped relationships of sleep with fitness or adiposity, but did pinpoint a trade-off between the adverse associations of shorter sleep with worse adiposity versus longer sleep with worse fitness. Our optimal sleep duration of 10.2 h aligns with international 24-h movement guidelines of 9–11 h [42–44].

Little is known about the shape of relationships between children's sedentary time and health outcomes. Our findings suggest that a surprisingly substantial proportion of the day (41%, 9.9 h/d) should be spent sedentary for best fitness/adiposity. Given the sample's minimum sedentary duration (7.8 h), it remains entirely possible that much lower levels of sedentary time could be beneficial, especially for fitness. Also, the range of sedentary durations associated with the best (overlapping) fitness/adiposity zone was quite wide (2 h 20 min, Fig 2) suggesting variability in estimates.

## Strengths and limitations

Strengths of this study include the large, population-based sample, objective measurements with high reliability and validity, and ability to include all daily activities in the same analysis. Our novel application of CoDA models (and allowing inclusion of non-linear quadratic relationships) enabled us to generate the first evidence for optimal daily time use across two important child health outcomes (fitness and adiposity).

Limitations include the cross-sectional design, precluding inference of causation or directionality. Although we talk about optimal days and durations, we cannot imply that achieving these activity durations will bring about improvements in outcomes. While adjusting for socio-demographic covariates, our findings may not fully generalize to the most disadvantaged children, and residual confounding (e.g., diet) remains possible. Body composition was measured in a semi-fasted state, thus small differences may have been due to timing of the participant's latest meal. It should be considered that $VO_{2max}$ was normalized for body mass, meaning it was more a measure of functional capacity rather than a measure of aerobic power. We adjusted fitness models for zBMI to partial out potentially confounding adiposity associations. However, conceptually it may not make sense to separate fitness from adiposity, especially because body mass is included in the derivation of $VO_{2max}$. Our optimal activity durations are directly dependent on the type of accelerometers, cut points and analytical algorithms used in the study. Accelerometer counts were collapsed into 60-second epochs, which is long in comparison to other paediatric studies, and may have contributed to relatively low estimates for MVPA in this study. Particularly, the use of fixed intensity cut-offs to classify intensity of physical activity may have introduced bias, especially for fitness models. This is because children with better fitness may achieve fixed thresholds with less effort than children with poorer fitness, potentially underestimating their time in MVPA [47]. In addition, the validation study used to create the cut-offs used metabolic equivalents (METs) which are inherently confounded by adiposity [48]. Thus, different decisions made during the accelerometry processing may lead to different daily activity estimates and different predicted optimal activity durations. Research using other datasets, cut-points and accelerometers is encouraged to validate or challenge our findings.

## Interpretation and implications

Time spent in movement behaviors clearly influences a wide range of health outcomes in both children and adults. Unfortunately, it is likely that different activity compositions are ideal for different outcomes, so that optimizing overall health will involve compromises. Fuligni [49], for example, found that the amount of sleep associated with the lowest levels of internalizing and

externalizing symptoms was more than one hour greater than the amount associated with the best academic performance in 15-year-olds. In our study we saw both consistencies and differences in the optimal activity composition for fitness and adiposity. Relationships and optima were similar for MVPA and sedentary time, but dramatically different for LPA and sleep. Possibly LPA contributes to better fitness via opportunities for physical conditioning, while sleep reduces adiposity through hormonal and/or metabolic mechanisms [50]. Were we to consider more outcomes (such as educational achievement or mental health), yet more optima would likely come into play, and the compromise optimum—the "Goldilocks Day"—would involve more complex trade-offs. Where there are multiple outcomes, some kind of weighting function may be needed to represent personal preferences and/or public health priorities.

## Future directions

Two big unknowns are the feasibility of achieving optimal durations, and to what extent changing activity *causes* change in outcomes. Prospective observational and intervention studies are needed, noting that experimentally achieving large changes in daily activity is very challenging. Although we truncated the empirical footprint at ±3SD of individual activities, some regions may be sparsely represented by samples of real children. Perhaps we should restrict the empirical footprint within tighter bounds using kernel density or certainty intervals or reframe activity guidelines as "target compositions to work towards" instead of prescriptive durations.

We took lower adiposity to be better because our lowest predictions were within healthy limits. Excessively low values indicating underweight would not be the best outcome, and future studies may consider defining optimal desirability bands.

We operationalized daily time use as a weighted average of weekdays and weekend days. Future work may differentiate between weekdays and weekend days or describe "the Goldilocks Week" rather than "the Goldilocks Day", or simply define optimal activities for school weeks vs holiday periods. There may be little room to modify time use on school days due to the constraints imposed by curricular activities, whereas weekend days could provide an opportunity to "catch up" on sleep and physical activity, while simultaneously avoiding sedentary behaviors.

Our study collapsed moderate and vigorous intensities of physical activity into MVPA because MVPA is relevant to current 24-hour movement behavior guidelines. However, some studies have found differential health associations for moderate and vigorous physical activity, for example, vigorous physical activity appears more beneficially associated with cardiorespiratory fitness than moderate physical activity [51]. It would be of interest for future studies to explore the two intensities separately.

## Conclusion

Optimal time use for children's fitness and adiposity involves trade-offs. To best balance *both* outcomes, estimated activity durations for sleep and LPA align with, but for MVPA may exceed, international recommendations. More definitive decisions about optimal time use should consider a wider range of health and wellbeing outcomes. Optimal time use will likely involve trade-offs between activities depending on value decisions, which is not captured by current one-size-fits-all guidelines.

## Supporting information

**S1 File. Balancing time use for fitness and adiposity: Evidence to inform 24-hour guidelines.** Analytical method.
(PDF)

**S2 File. Results for individual fitness and adiposity measures.**
(PDF)

**S3 File. Interactive 3-D plot.**
(HTML)

## Acknowledgments

This paper uses unit record data from Growing Up in Australia, the Longitudinal Study of Australian Children. The study is conducted in partnership between the Department of Social Services (DSS), the Australian Institute of Family Studies (AIFS) and the Australian Bureau of Statistics (ABS). The findings and views reported in this paper are solely those of the authors and should not be attributed to DSS, AIFS or the ABS. DD had full access to all the data in the study and takes responsibility for the integrity of the data and the accuracy of the data analysis. REDCap (Research Electronic Data Capture) electronic data capture tools were used in this study. More information about this software can be found at: www.project-redcap.org. We thank the LSAC and CheckPoint study participants, staff and students for their contributions.

## Author Contributions

**Conceptualization:** Dorothea Dumuid, Melissa Wake, Timothy Olds.

**Data curation:** Melissa Wake, David Burgner, Ben Edwards, Timothy Olds.

**Formal analysis:** Dorothea Dumuid.

**Funding acquisition:** Melissa Wake, David Burgner, Terence Dwyer, Timothy Olds.

**Investigation:** Dorothea Dumuid, Melissa Wake, David Burgner, Mark S. Tremblay, Anthony D. Okely, Ben Edwards, Terence Dwyer, Timothy Olds.

**Methodology:** Dorothea Dumuid, David Burgner, Mark S. Tremblay, Anthony D. Okely, Ben Edwards, Terence Dwyer, Timothy Olds.

**Project administration:** Melissa Wake.

**Resources:** Melissa Wake.

**Visualization:** Dorothea Dumuid, Timothy Olds.

**Writing – original draft:** Dorothea Dumuid, Melissa Wake, Timothy Olds.

**Writing – review & editing:** Dorothea Dumuid, Melissa Wake, David Burgner, Mark S. Tremblay, Anthony D. Okely, Ben Edwards, Terence Dwyer, Timothy Olds.

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
