## [Decision Letter · Decision Letter 0]

16 Oct 2020

PONE-D-20-20393

Balancing time use for children’s fitness and adiposity: evidence to inform 24-hour guidelines for sleep, sedentary time and physical activity.

PLOS ONE

Dear Dr. Dumuid,

Thank you for submitting your manuscript to PLOS ONE. After careful consideration, we feel that it has merit but does not fully meet PLOS ONE’s publication criteria as it currently stands. Therefore, we invite you to submit a revised version of the manuscript that addresses the points raised during the review process.

In particular there are two areas that influence on your result that need further attention. Firstly, as mentioned by reviewer two the association between fitness (i.e. VO2) and fatness (BMI) is by default strongly associated. This is because both of them carry identical information, namely weight (Kg), VO2 is expressed as mlO2/kg and BMI as kg/m2. In addition the association between height (m) and O2 are also quite high (reports of r~0.7-0.8). Thus it can be said that those variables are providing almost identical information into the models thus causing collinearity which hampers the interpretation of the models.  

Secondly, there are a lot of information avaliable regarding the missuse of compositional data analysis and associated methods when analysing accelerometer data and a discussion regarding how you avoided the pitfalls as outlined in Aadland E, Kvalheim OM, Anderssen SA, Resaland GK, Andersen LB. Multicollinear physical activity accelerometry data and associations to cardiometabolic health: challenges, pitfalls, and potential solutions. Int J Behav Nutr Phys Act. 2019 Aug 27;16(1):74. is needed.

We look forward to receiving your revised manuscript.

Kind regards,

Patrick Bergman

Academic Editor

PLOS ONE

Journal Requirements:

2. In your Methods section, please provide additional details regarding the recruitment of participants to this study, including any inclusion and exclusion criteria.

3.We note that you have indicated that data from this study are available upon request. PLOS only allows data to be available upon request if there are legal or ethical restrictions on sharing data publicly. For information on unacceptable data access restrictions, please see http://journals.plos.org/plosone/s/data-availability#loc-unacceptable-data-access-restrictions.

Reviewers' comments:

Reviewer's Responses to Questions

**Comments to the Author**

1. Is the manuscript technically sound, and do the data support the conclusions?

Reviewer #1: Yes

Reviewer #2: Yes

2. Has the statistical analysis been performed appropriately and rigorously? 

Reviewer #1: Yes

Reviewer #2: Yes

3. Have the authors made all data underlying the findings in their manuscript fully available?

Reviewer #1: Yes

Reviewer #2: No

4. Is the manuscript presented in an intelligible fashion and written in standard English?

Reviewer #1: Yes

Reviewer #2: Yes

5. Review Comments to the Author

Reviewer #1: The manuscript ” Balancing time use for children’s fitness and adiposity: evidence to inform 24-hour guidelines for sleep, sedentary time and physical activity” by Dumuid et al. is interesting and nicely conducted study on the links between sleep, sedentary time, and physical activity with adiposity and fitness in a large cross-sectional sample of children. The manuscript is an important addition to the literature on possible health benefits of 24h movement guidelines. The manuscript is also well written and easy to follow. However, I have some important points for the Authors to consider.

Major comments

Assessment of physical activity and the use of accelerometry data. The Authors present amounts of LPA and MVPA in relation to adiposity and fitness outcomes. I think it is important that the Authors consider that any fixed intensity cut-off would cause significant miss-classification of PA intensity. This is especially important when one outcome is physical fitness. It is plausible that children with better fitness levels achieve those fixed thresholds with less effort than lower with children and therefore underestimate their time spent in MVPA (e.g. PMID: 32620888). This is crucial point as miss-classification PA intensity would cause remarkable bias to the results. In addition, the PA measure may be a source of bias because the validation study used to create the used the intensity cut-offs utilised MET-values in creating the cut-offs. METs are inherently confounded by adiposity (PMID: 26321958) and fixed MET-based cut-offs cause misclassification of PA intensity (see PMID: 32620888). Of note, MET cut-offs 3-4 and 6-7 are often used, but they lack reasonable physiological rationale. Furthermore, in addition to possible misclassification of PA intensity, the results would markedly differ in any other accelerometry cut-points would have been used (PMID: 30548545). I acknowledge that currently there are very limited possibilities for individual calibration of accelerometry cut-points, but keeping this information in mind I suggest that the authors more carefully interpret the results and include some discussion about this perhaps into limitations section.

Can the authors provide some reason for 60 s epoch? It is rather long for paediatric studies.

Vigorous intensity PA has been considered important for health outcomes and based on evidence for intervention studies it would be more important for cardiorespiratory fitness than moderate intensity exercise. VPA is also included the PA recommendations by WHO and PA recommendations for Americans. Thus, I suggest that the Authors also include VPA into the manuscript.

I also suggest that the Authors interpret the cardiorespiratory fitness data carefully as VO2peak normalised for body mass is not a pure measure of aerobic power but also adiposity making it more a measure of functional capacity than physiological measure of the function of cardiovascular and skeletal muscle systems (please see e.g. reviews by Welsman and Armsntrong). Thus, I suggest that the Authors also highlight this limitation and discuss how it may affect on the results.

Specific comments

1. Introduction

The introduction would benefit more detailed description why each component (incl. VPA) would be associated with adiposity and fitness.

2. Methods

L99. How sleep was derived from the accelerometry data? I assume that sleep was treated as more is better? Please also consider that more is not always better.

Assessment of body composition. Were bioimpedance analysis taken in fasted stated or how was it standardised? Please explain. Please also explain why body composition was expressed at it was. To enhance the comparability between studies the Authors could also report BF% in the supplement.

Does composite adiposity score add anything to this paper. What the Authors think that it describes?

3. Discussion

Please adjust the discussion according to previous comments.

Please be careful with any causal terminology as this was cross-sectional study.

L270. Please be careful in expressing volume number true because thresholds and methods used to assess PA will have effect on those estimates.

Reviewer #2: The authors of this manuscript are well establisehd in conducting state of the art analyses of 24h movement behaviours. The current literature comprises many studies using compositional data analysis to investigate the effect of 24h movement behaviours on diffferent health outcomes. The present manuscript describes the analysis of optimal compositions of 4 main behaviours with regard to two important health outcomes, i.e. fitness and adiposity, to provide recommendations with regard to the interrelation of behaviours. Overall, the manuscript shows a high quality and concludes interesting results using this new analysis. Thus, I would reommend to accept this manuscript for publication after some minor revisions.

Line 63 and 65: The wording adiposity seems quite confusing in combination with 'better fitness' or 'beneficial'. Please use lower weight status or similar.

Line 76f.: The authors might include more information on prior work about compositional data analysis.

Line 85-90: Some information about the sample size from Fig 1 might be included in the text.

Line 106: A short description of the algorithms (at least concept and main parameters) for sleep detection and non-wear time (similar to non-wear detection by Choi?) should be included for the reader.

Line 172: Please state the covariates that are used for adjustment.

Line 215: Again the use of adiposity. Best mean adiposity should be actually a mean weight status, right? high weight status refers to adiposity but optimal adipositiy seems to be average weight status? Please correct precisely.

Line 239: Please discuss the use of the term adiposity measure, since you actually use weight status.

Line 327: The term lower adiposity actually should refer to zBMI, i.e. weight status. Please reconsider the wording.

6. PLOS authors have the option to publish the peer review history of their article (what does this mean?). If published, this will include your full peer review and any attached files.

Reviewer #1: No

Reviewer #2: No

---

## [Author Response · Author response to Decision Letter 0]

12 Nov 2020

Many thanks to the editor and reviewers for the constructive comments on our manuscript. Please see the "Response to Reviewer" document for a point-by-point response to the comments, and please see our revised manuscript which we hope now meets the requirements for publication in PLOS ONE.

Regards,

Dorothea Dumuid.

---

## [Decision Letter · Decision Letter 1]

4 Jan 2021

Balancing time use for children’s fitness and adiposity: evidence to inform 24-hour guidelines for sleep, sedentary time and physical activity.

PONE-D-20-20393R1

Dear Dr. Dumuid,

We’re pleased to inform you that your manuscript has been judged scientifically suitable for publication and will be formally accepted for publication once it meets all outstanding technical requirements.

Kind regards,

Patrick Bergman

Academic Editor

PLOS ONE

Additional Editor Comments (optional):

Reviewers' comments:

Reviewer's Responses to Questions

**Comments to the Author**

1. If the authors have adequately addressed your comments raised in a previous round of review and you feel that this manuscript is now acceptable for publication, you may indicate that here to bypass the “Comments to the Author” section, enter your conflict of interest statement in the “Confidential to Editor” section, and submit your "Accept" recommendation.

Reviewer #2: All comments have been addressed

Reviewer #3: (No Response)

2. Is the manuscript technically sound, and do the data support the conclusions?

Reviewer #2: Yes

Reviewer #3: Yes

3. Has the statistical analysis been performed appropriately and rigorously? 

Reviewer #2: Yes

Reviewer #3: Yes

4. Have the authors made all data underlying the findings in their manuscript fully available?

Reviewer #2: No

Reviewer #3: Yes

5. Is the manuscript presented in an intelligible fashion and written in standard English?

Reviewer #2: Yes

Reviewer #3: Yes

6. Review Comments to the Author

Reviewer #2: (No Response)

Reviewer #3: Review Comments to the Author

Please use the space provided to explain your answers to the questions above. You may also include additional comments for the author, including concerns about dual publication, research ethics, or publication ethics. (Please upload your review as an attachment if it exceeds 20,000 characters) (Limit 100 to 20000 Characters)

Accept manuscript

7. PLOS authors have the option to publish the peer review history of their article (what does this mean?). If published, this will include your full peer review and any attached files.

Reviewer #2: No

Reviewer #3: No

---

## [Editor Report · Acceptance letter]

7 Jan 2021

PONE-D-20-20393R1 

Balancing time use for children’s fitness and adiposity: evidence to inform 24-hour guidelines for sleep, sedentary time and physical activity. 

Dear Dr. Dumuid:

I'm pleased to inform you that your manuscript has been deemed suitable for publication in PLOS ONE. Congratulations! Your manuscript is now with our production department. 

Kind regards, 

on behalf of

Dr. Patrick Bergman 

Academic Editor

PLOS ONE